# Dextran Sulfate Nanocarriers: Design, Strategies and Biomedical Applications

**DOI:** 10.3390/ijms24010355

**Published:** 2022-12-26

**Authors:** Subramaniyan Ramasundaram, Gurusamy Saravanakumar, Sivasangu Sobha, Tae Hwan Oh

**Affiliations:** 1School of Chemical Engineering, Yeungnam University, Gyeongsan 38436, Republic of Korea; 2OmniaMed Co., Ltd., Pohang 37666, Republic of Korea

**Keywords:** dextran sulfate, nanomaterials, nanocarrier, electrostatic interactions, drug delivery, imaging, polymer therapeutic

## Abstract

Dextran sulfate (DXS) is a hydrophilic, non-toxic, biodegradable, biocompatible and safe biopolymer. These biomedically relevant characteristics make DXS a promising building block in the development of nanocarrier systems for several biomedical applications, including imaging and drug delivery. DXS polyanion can bind with metal oxide nanomaterials, biological receptors and therapeutic drug molecules. By taking advantage of these intriguing properties, DXS is used to functionalize or construct nanocarriers for specific applications. In particular, the diagnostic or therapeutic active agent-loaded DXS nanoparticles are prepared by simple coating, formation of polyelectrolyte complexes with other positively charged polymers or through self-assembly of amphiphilic DXS derivatives. These nanoparticles show a potential to localize the active agents at the pathological site and minimize undesired side effects. As DXS can recognize and be taken up by macrophage surface receptors, it is also used as a targeting ligand for drug delivery. Besides as a nanocarrier scaffold material, DXS has intrinsic therapeutic potential. DXS binds to thrombin, acts as an anticoagulant and exhibits an inhibitory effect against coagulation, retrovirus, scrapie virus and human immunodeficiency virus (HIV). Herein, biomedical applications involving the use of DXS as nanocarriers for drugs, biomolecules, and imaging agents have been reviewed. A special focus has been made on strategies used for loading and delivering of drugs and biomolecules meant for treating several diseases, including cancer, inflammatory diseases and ocular disease.

## 1. Introduction

Dextran sulfate (DXS) is a hydrophilic, biodegradable, biocompatible and negatively charged polysaccharide. It is a highly branched polyanionic polysaccharide with a sulfur content of about 17%, which is approximately equivalent to ~2.3 sulfate groups per glucose unit (Figure 1). This sulfated polysaccharide is produced by the esterification of dextran with chlorosulphonic acid [1,2,3,4]. The parenteral polymer of DXS, dextran, is naturally produced by lactic acid based Leuconostoc mesenteroides bacterial species [5]. DXS has versatile physicochemical and biological properties. As such, DXS possesses many favorable intrinsic characteristics. For example, it has been shown to have antiviral, antibacterial, anti-inflammatory, antifibrotic, and wound-healing properties [6,7]. It has also shown a lipemia-clearing effect and is clinically used for treating high blood lipid levels [8]. The negatively charged DXS can bind thrombin and prevent the coagulation of blood. [9]. DXS can act as a stabilizer and prevents the heat-induced aggregation of proteins [10,11]. In addition, it is also used as an additive in cell culture media for preventing cell aggregation and in cosmetics as a gel-forming agent [12,13,14].

From a pharmaceutical standpoint, DXS holds several advantages as a primary constituent of drug carriers [1,15,16]. The sodium salt of DXS is highly water-soluble and non-toxic. In particular, the intrinsic negatively charged DXS can readily form complexes with positively charged drug molecules, biomolecules (such as genes, proteins and growth factors) and biopolymers (such as chitosan and its derivatives) in mild conditions, and thus a facile nano-sized drug delivery platform can be created for the delivery of a wide range of therapeutic agents [17,18,19,20,21]. The surface characteristics of these carriers can be fine-tuned or controlled by varying the content of DXS. Moreover, DXS can also be chemically conjugated with drug molecules or hydrophobic moieties to form polymeric prodrug or amphiphilic DXS derivatives that can be self-assembled into nanoparticles in an aqueous condition. These self-assembled DXS-based nanostructures enable the delivery of poorly water-soluble drugs. More importantly, DXS can bind to scavenger receptor class A (SR-A), which is one of the several receptors overexpressed on the activated macrophages in inflammation sites, and therefore DXS-based nanoparticles have huge potential for the selective and targeted delivery of anti-inflammatory drugs to inflammatory sites [22,23].

In this review, the design strategies for the preparation of DXS nanoparticles and their biomedical applications, such as drug delivery and imaging, are discussed. This review provides an overview and discusses the DXS-based nanoparticles meant for delivering drugs and therapeutic biomolecules for various diseases, including cancer, inflammatory diseases such as rheumatoid arthritis (RA) and ocular disease. Furthermore, it highlights DXS-based nanoplatforms for the target-specific imaging of inflammatory diseases.

## 2. Strategies for the Preparation of DXS-Based Nanoparticles

Based on the solubility, charge, and other intrinsic characteristics of the biopolymers, several methods have been developed for the preparation of nanoparticles. A number of papers have reviewed these methods in detail [24,25,26]. In general, DXS-based nanoparticles are mostly prepared by the following three methods (Figure 2).

### 2.1. DXS-Drug Nanocomplexes

Owing to its high negative charge density, DXS can readily form nanocomplexes with positively charged ions or small molecule drugs by simple mixing through electrostatic interactions (Figure 2A). The formation of ionic bonds between the DXS polymer chains by the drug molecules, in combination with the hydrogen bonds and other hydrophobic interactions of drugs, is the main mechanism involved in the formation of nanocomplexes. Several factors, such as the size of the drug and the weight ratio of drugs to DXS, influence the formation of DXS–drug nanocomplexes. As the electrostatic interactions between the DXS and drugs are sensitive to pH, the loaded drug can be designed to be released in response to pH changes at the targeted pathophysiological sites. However, this pH sensitivity can also induce instability of the ionically crosslinked network. Nonetheless, mild preparation conditions without the aid of other toxic organic solvents or other constituents make this strategy appealing for drug delivery applications.

### 2.2. DXS-Based Polyelectrolyte Complex (PEC) Nanoparticles

Generally, PEC nanoparticles are prepared by mixing oppositely charged polyelectrolyte solution in non-stochiometric ratios (Figure 2B). The interest in the formation of soluble complexes between oppositely charged polyelectrolytes has gained significant impetus since the work of Tsuchida, Kabanov, and Fukuda [27,28,29]. Because of its negatively charged sulfate groups, DXS polyanion can readily interact with other polycations to form PEC nanoparticles, which have been explored for various pharmaceutical applications, including drug and gene delivery systems. DXS can form complexes with several kinds of polycations, such as cationic polysaccharides, cationic synthetic polymers, and cationic synthetic polypeptides [1]. In particular, positively charged chitosan (CH) has been widely used as a polycation for the preparation of DXS-based PEC nanoparticles [30]. Since no toxic crosslinkers are involved in this method, PEC nanoparticles are considered to be biocompatible, which is one of the important prerequisites for biomedical applications. As the formation of PEC is largely dependent on the interactions between the polyelectrolytes, the size and the surface characteristics of the PEC are influenced by several parameters, including concentration, molecular weight and charge density of the polyelectrolytes. Besides these parameters, other factors such as the ionic strength, pH of the solution, temperature, and mixing order may also influence the formation of stable PEC. The stability of DXS nanoparticles in biological fluids is crucial for effective drug delivery and to improve the in vivo therapeutic efficacy of drugs [31]. In recent years, covalently crosslinking of the core or shell of polymeric nanoparticles has received increasing attention to improve their in vivo stability. Guarino et al. improved the DXS-CH PEC nanoparticles by covalently crosslinking the core using a short-chain dicarboxylic acid (succinate) without affecting the shell of the nanoparticles [32]. These core-crosslinked nanoparticles were stable in NaCl solutions up to 3 M. Furthermore; these nanoparticles could maintain the stability and biological activity of loaded heparin-binding proteins. When these crosslinked nanoparticles were incorporated with stromal cell derived factor 1α (SDF-1α) and delivered to the lungs of rats as aerosol, they showed a 17-fold longer retention in vivo than that of free protein.

### 2.3. Self-Assembled DXS-Based Nanoparticles

In an aqueous condition, polymeric amphiphiles can form self-assembled nanoparticles in which the inner core is hydrophobic and outer shell is hydrophilic (Figure 2C) [33,34]. Amphiphilic DXS derivative can be synthesized by conjugating hydrophobic constituents to the hydroxy functional group at the backbone or at the reducible chain end of DXS. This self-assembly process is driven by hydrophobic interactions in order to minimize interfacial free energy. The hydrophobic inner cores can be used for the delivery of hydrophobic therapeutic drugs or imaging agents. The hydrophilic DXS shells provide prolonged circulation and can act as targeting ligands for cell-specific targeted delivery (SR-A mediated activated macrophage targeting). Furthermore, these nanoparticles can be labeled with fluorescent dye molecules for in vivo bioimaging. The size and surface characteristics of these nanoparticles can be controlled by varying the degree of substitution of hydrophobic moieties and/or the size of the polymer chain [35]. It should also be noted that the excess hydrophobic modification of DXS may result in the precipitation of DXS amphiphiles under aqueous conditions. At the same time, a relatively very low amount of hydrophobic constituent may not induce particle formation or result in unstable loose particles. Thus, an optimal degree of substitution should be maintained to obtain particles with desired size and stability for improved in vivo performances.

## 3. Therapeutic Applications of DXS Nanoparticles

### 3.1. Cancer

Most anticancer drugs exhibit poor water solubility. Overall, it has been estimated that about 75% of new drug candidates that are currently under development in the pharmaceutical pipelines have poor water solubility, and the majority of these are anticancer drugs [36,37]. The low solubility of these drugs, in combination with other factors, leads to poor pharmacokinetic performance and thus severely limits their application in cancer therapy. As the majority of these drugs are weak acids or bases, the preparation of polymer-drug nanocomplexes by electrostatic interactions between the charged drug molecules with oppositely charged biopolymers is a promising approach. In this context, owing to high charge density and chain flexibility, DXS has promising potential to form DXS-drug nanocomplexes with cationic drug molecules. For example, Yousefpour et al. prepared nano-sized drug delivery systems by simple complexation of DXS and cationic anticancer drug doxorubicin (DOX) [38]. The primary amino group of the DOX molecule (with pKa of 8.6) bestows positive charge characteristics and enables the formation of DXS-DOX nanocomplexes at a DXS/DOX weight ratio of 0.4–0.6. They found that electrostatic hydrogen bonding and π-π stacking interactions among the DOX molecules play an important role in the complex formation. The nanocomplexes showed an initial fast DOX release followed by a slow and sustained release. Alternatively, DOX molecules are also encapsulated into PEC nanoparticles prepared using DXS and positively charged CH [39]. In this method, the first positive charge of the DOX was neutralized by complexation with DXS, followed by the addition of CH solution. Finally, the cross-linking of the CH was initiated by the addition of ionic crosslinker sodium tripolyphosphate (TPP). Even at 10% (*w*/*w*) of DXS with respect to CH, the DOX loading content was increased 2-fold compared to control formulations without or with other polyanions. In vitro confocal studies indicated that these DXS-CH PEC nanoparticles could enter the cells via an endocytic mechanism and release DOX intracellularly. A similar approach, without TPP crosslinking, was used to encapsulate hydrophobic curcumin into DXS-CH PEC nanoparticles [40]. Curcumin is a small molecule phytodrug that exhibits a variety of biological activities and has been shown to be effective against many cancers. Owing to its poor solubility, it faces several issues, such as poor bioavailability and rapid metabolism. Curcumin-loaded nanoparticles showed good colloidal stability and were toxic to cancer cells. When compared to the O-carboxymethyl chitosan nanoparticles, DXS-CH PEC nanoparticles showed a better drug release profile, about 70% curcumin was released after 120 h. Besides simple PEC nanoparticles, polyelectrolyte multilayer nanoparticles have also been developed for dual drug delivery. By employing DXS and CH as polyanions and polycations, respectively, Wang et al. developed three-layer (CH/DXS/CS) PEC nanoparticles via a layer-by-layer (LBL) electrostatic assembly technique (Figure 3) [41]. The inner core of the nanoparticle is encapsulated with the hydrophobic anticancer drug paclitaxel (PTX), and the outer layer is accommodated with the hydrophilic drug 5-fluorouracil (5-Fu). The drug release rate was accelerated in acidic media. The 5-Fu release is more likely to be Fickian diffusion, while the PTX release corresponds to a combination of diffusion and erosion mechanisms. The therapeutic mechanisms of these two are also different. While PTX disrupts the dynamics of microtubules and induces mitotic arrest that can eventually lead to cell death, the 5-Flu inhibits DNA synthesis by inhibiting thymidine nucleotide synthase. Combining these two drugs into a single platform may have a chance to inhibit the resistance of cancer cells to anticancer drugs by killing cancer cells at different growth stages. The cytotoxicity of nanoparticles was studied using HepG2 cancer cells. These dual drug-loaded nanoparticles internalized into HepG2 cells and exhibited synergistic cytotoxic effects against the cancer cells.

In addition to enhancing the pharmacokinetics of poorly water-soluble cytotoxic drugs, several DXS-based nanoparticles have also been developed for the delivery of hydrophilic anticancer drugs. With the aid of DXS, vincristine (VC) sulfate, a freely water-soluble anticancer drug, was encapsulated into cetyl palmitate solid lipid nanoparticles using the microemulsion method [42]. By this method, the encapsulation efficiency of VC reached up to 93%, and the drug release profile of the nanoparticles indicated sustained VC release. Further, VC-loaded nanoparticles exhibited a comparable cytotoxicity effect to that of VC sulfate solution against MDA-MB-231 cells. When injected into male Wistar rats, VC-loaded nanoparticles showed higher plasma and tissue concentration and longer drug mean residence time compared to the free VC solution. Studies performed using coumarin-6 as a model drug revealed that the use of DXS-based nanoparticle carriers can increase drug delivery to the brain by almost five times. Multidrug resistance (MDR) is a major bottleneck limiting the potential of cancer chemotherapy. Therefore, significant research efforts have been focused on overcoming MDR in cancer. Nanoparticle-based drug delivery holds huge potential to deliver chemotherapeutic agents for the treatment of MDR cancer [43,44]. To alleviate MDR, lipid-DXS-based hybrid nanocarriers were developed and loaded with water-soluble cationic mitoxantrone hydrochloride (MTO) [45]. Similar to the above system, these lipid-based DXS nanoparticles also showed enhanced drug loading efficiency (97.4%) and showed a sustained release profile. The pharmacokinetics studies performed in rats demonstrated that nanoparticles had a longer half-life than MTO solution. The DXS-hybrid nanoparticles also improved the biodistribution of MTO in plasma, spleen and brain. These nanoparticles also entered into breast cancer resistance (BCR)-overexpressing MCF-7/MX cells by the clathrin-mediated endocytosis pathway and thus overcoming the MDR of MCR-7/MX cells. These results indicate that this hybrid DXS-based carrier system might have the potential to overcome MDR in chemotherapy.

### 3.2. Rheumatoid Arthritis (RA)

RA is the most commonly occurring, progressive, autoimmune disease. It is characterized by inflammation of the synovium, cartilage and bone, leading to the destruction of joint tissue, swelling and dysfunction [46,47,48]. There are four general classes of drugs commonly used for treating RA: non-steroidal anti-inflammatory drugs (NSAIDs), glucocorticoids, disease-modifying anti-rheumatic drugs (DMARDs) and biologics [49]. Owing to poor bioavailability and short biological half-life, RA therapy requires high and frequent dosing. Moreover, the poor target selectivity of these drugs also causes severe side effects and extra-articular manifestations. Thus, achieving targeted and controlled delivery of drugs for arthritis is crucial to improve the treatment of RA. The activated macrophage plays an important role in the pathophysiological process of RA [50]. Since activated macrophages are over-expressed with SR-A, which can be recognized and bound by DXS, DXS-based nanoparticles can serve as promising target-specific carrier systems for the treatment of RA.

Yang et al. prepared a DXS-graft-methotrexate (DXS-b-MTX) conjugated prodrug system by chemically conjugating the drug MTX, which is a first-line DMARD agent for most patients with RA, to DXS [51]. Because of the amphiphilic characteristics, this prodrug conjugate was self-assembled into micelles in an aqueous solution. The presence of DXS allowed the selective targeting of SR-A macrophages associated with RA. After intravenous injection, these prodrug micelles preferentially accumulate in the inflamed area more than the normal cells. In particular, these DXS-g-MTX micelles inhibited pro-inflammatory cytokines, alleviated synovitis, and protected the articular cartilage compared to the control micelles prepared using non-sulfated dextran. In another strategy, MTX was physically loaded into layered double hydroxide nanocomposites and subsequently coated with DXS [52]. Since LDH exhibits pH sensitivity, these nanocomposites exhibited slightly accelerated MTX release at acidic conditions. As the synovial tissue of patients with RA is in a low-pH environment and positively correlates with the severity of the disease, the pH-responsive drug release is beneficial for treating RA. The nanocomposites showed enhanced cellular uptake compared to that of the non-targeting carrier. In vivo pharmacodynamic studies demonstrated improved therapeutic efficacy in adjuvant-induced arthritis rats compared with the free MTX alone. In addition to the acidic condition, enzyme matrix metalloproteinase 2 (MMP-2) is excessively secreted in the joints of patients with RA and plays a crucial role in inflammation and immunity. To develop an MMP-2 enzyme-sensitive drug delivery nanoplatform, Yu et al. conjugated Celastrol (Cel), a pentacyclic triterpene compound that has shown the ability to inhibit the development of RA, to the DXS via an MMP-2 sensitive peptide (PVGLIG) (Figure 4) [53]. The resulting DXS-PVGLIG-Cel conjugated formed self-assembled prodrug micelles with an average size of about 190 nm and negative surface charge. These micelles had a high entrapment efficiency (about 44%). Further, the micelles showed good sensitivity to the MMP-2 enzyme and released about 78% of the loaded drug in the in vitro release medium containing the MMP-2 enzyme, suggesting that micelles can effectively deliver Cel to the activated macrophages. Compared to the free Cel, the prodrug micelles showed a better anti-rheumatoid arthritis effect.

### 3.3. Ocular Disease

As the eye has several defense systems and physiological barriers, effective delivery of drugs to the eye remains a great challenge. The low bioavailability of ocular delivery systems is largely due to the poor permeation and rapid clearance from the eye after administration [54]. In recent years, biodegradable nanoparticle-based formulations have greatly improved the retention time, bioavailability, and controlled release via mucoadhesion to the epithelia in the eye. For ocular drug delivery, DXS-CH PEC nanoparticles are widely employed because of their mucoadhesive characteristics. Chaiyasan et al. prepared DXS-CH nanoparticles and explored their potential for the topical ocular delivery of the drug lutein, which is used for preventing cataracts and vision loss in elderly adults [55]. These PEC nanoparticles were further crosslinked and stabilized using a carbodiimide-based activating agent and polyethylene glycol, respectively. By controlling the feed ratios, the surface charge of the PEC nanoparticles was tuned to be positive and thus found to be mucoadhesive. Compared to free form, the lutein loaded in the DXS-CH nanoparticles showed long storage stability. All these characteristics make the nanoparticle formulation suitable for delivery to the ocular surface. The DXS-CH PEC nanoparticles also exhibited good stability to lysozyme, which is found in tears [56]. This high stability confers prolonged adherence of nanoparticles to the corneal surface, facilitating controlled and sustained drug delivery. The superficial epithelial layers with a number of tight junctions restrict the penetration of hydrophilic drugs. However, the leaky corneal endothelial monolayer allows the penetration of macromolecules. Therefore, after labeling the DXS-CH PEC nanoparticles with fluorescence isothiocyanate, their mucoadhesive and penetration characteristics were studied [57]. These positively charged nanoparticles with a size of about 400 nm were able to retain on the porcine ocular surface for a duration longer than 4 h and partially penetrated in the corneal epithelium, suggesting that DXS-CH PEC nanoparticles could be useful for sustained drug delivery to the ocular surface and epithelium. DXS-CH PEC nanoparticles are also loaded with the drug antibiotic drug ciprofloxacin (Cipro), which is effective against various microorganisms [58]. A fine crosslinking between DXS and CH rendered a monotonous controlled release of the drug for 21 h. The Cipro-loaded nanoparticles were tested against commonly known Gram-positive and Gram-negative microorganisms causing eye infections. The antimicrobial efficacy of the release of Cipro remained stable even after 24 h. Instead of polycationic CH, DXS nanocomplexes are prepared by complexing an amide-type analgesic drug bupivacaine (BUP) with different polyanionic materials (DXS, carboxymethylcellulose and carboxymethyl dextran), followed by crosslinking using calcium ion as the ionic crosslinker [59]. The optimized nanocomplexes with BUP, DXS, and calcium ion at the ratio of 1:20:20 (*w*/*w*/*w*) showed a sustained release profile over 7 days. Therefore, these nanocomplexes were considered effective long-active delivery systems for the analgesic agent.

## 4. Diagnostic Applications of DXS Nanoparticles

The advent of new imaging modalities has enabled the visualization and characterization of biological processes at the cellular and molecular levels. In particular, polymer or hybrid nanoparticle-based imaging agents are receiving increased attention and are utilized for generating high-resolution and high-contrast images for accurate and precise diagnostics [60,61,62]. Recently, DXS-based nanoparticles have been used as imaging agents for non-invasive targeted biomedical imaging. This section discusses the DXS-based nanoparticles for optical and magnetic resonance imaging (MRI)

### 4.1. DXS Nanoparticles for MRI

For better visualization, high-quality imaging and cellular-level monitoring, magnetic iron oxide nanoparticles have been utilized as contrast agents. The surface modification of these nanoparticles with DXS was performed to prevent aggregation, improve biocompatibility and increase blood circulation. In fact, non-sulfated dextran-coated superparamagnetic nanoparticles (SPIONS) have already been well-recognized as multifunctional imaging agents for MRI [63]. In addition, several dextrans or derivatized dextran-coated SPIONs have already been clinically approved or are currently in clinical trials [64]. As discussed earlier, DXS can be recognized and taken up by activated macrophages through macrophage surface receptors in the atherosclerotic plaques. Thus, DXS-coated SPIONs have huge potential as MRI contrast agents for cardiovascular imaging. Louie and co-workers synthesized DXS-coated SPION by alkaline coprecipitation method by using a combination of DXS and dextran [65]. By optimizing the polymer content, the ratio of iron salts and reaction time, they obtained DXS-coated SPIONs with good MR properties (r1 = 14.46 mM^−1^ s^−1^ and r2 = 72.55 mM^−1^ s^−1^) in a good yield. However, it was difficult to obtain monodisperse DXS-SPIONs using this procedure. To surmount this issue, they developed a new, improved method by sulfating dextran-coated SPION using sulfur trioxide (Figure 5A) [66]. The DXS-SPIONs obtained by this method had a hydrodynamic size of 62 nm, with r1 = 18.1 mM^−1^ s^−1^ and r2 = 95.8 mM^−1^ s^−1^ (37 °C, 1.4 T). In vitro cell studies demonstrated that these nanoparticles are non-toxic and specifically taken by macrophages via the receptor-mediated endocytosis process. In vivo MRI studies using an atherosclerotic mouse injury model also showed preferential uptake of the DXS-SPIONs at the site of atherosclerotic plaque compared to the non-sulfated analogues. In another study, SPIONs modified with DXS-CH PEC are prepared by in situ coating method, where SPIONS are coprecipitated within the PEC matrices through the direct addition of ammonia water [67]. Incubation of these DXS-CH-coated SPIONs with BALB/c 3T3 fibroblast cells did not alter the cell viability, confirming its biocompatibility. These DXS-CH-coated SPIONs were taken by 3T3 cells after overnight incubation and thus can be used as contrast agents for cell tracking. Although SPION can be readily modified with DXS-CH PEC layers, the formation and stability of PEC are determined by several factors. For effective in vivo MRI applications, a stable coating of DXS on the surface of SPION is crucial. To achieve a stable and robust coating of DXS on the SPION surface, Park and co-workers synthesized a double hydrophilic DXS-*b*-poly(glycerol methacrylate) (DXS-*b*-PGMA) (Figure 5B), where the DXS segment acts as a ligand for SR-A on activated macrophages, and PGMA segment with 1,2-diol moieties acts as a strong surface-anchoring component for SPIONs [68]. Similar to the previously discussed procedures, DXS-modified SPIONs are prepared by a simple coprecipitation method in the presence of a DXS-*b*-PGMA copolymer. Compared to the control dextran-coated SPIONs, DXS-SPIONs obtained using this method showed high aqueous stability and were taken by macrophages by SR-A mediated endocytosis. Further, it also produced a distinct contrast enhancement in the T_2_-weighted MR cellular imaging of activated macrophages, indicating its potential as a contrast agent for atherosclerosis imaging. However, additional in vivo experiments are needed to investigate the stability and efficiency of these nanoparticles for diagnostic applications.

### 4.2. DXS Nanoparticles for Optical Imaging

In vivo optical imaging is a non-invasive, safe and highly sensitive technique and can provide fast and real-time imaging [69,70]. This method utilizes light to probe cellular and molecular events in living subjects. However, conventional optical imaging suffers from undesired autofluorescence from biological tissues, scattering, and poor signal-to-background ratio, leading to reduced imaging sensitivity. The recent advances in sophisticated optical approaches, such as optical coherence tomography and diffuse optical tomography, have greatly minimized these limitations. To date, a number of optical imaging probes, including organic fluorophores and fluorescence semiconductor nanoparticles, have been reported [71,72]. In particular, these probes are easy to handle, lack radiation and can be tagged with other functional moieties. Near-infrared (NIR) fluorescence imaging is widely employed for in vivo small animal preclinical trials. Since DXS nanoparticles can be readily conjugated with NIR fluorophores and have the ability to target activated macrophages, they can be used as an optical nanoprobe for the diagnosis of a number of chronic inflammatory diseases. As activated macrophages are abundant in the inflamed joints of patients with RA, a self-assembled DXS nanoprobe was prepared by labeling DXS-*b*-poly(caprolactone) (DXS-b-PCL) copolymer with NIR dye Cyanine 5.5 (Cy5.5) [73]. The amphiphilic DXS-b-PCL copolymer was synthesized by chemically conjugating alkyn-end functionalized DXS and azide-end functionalized PCL via click chemistry. In vitro cellular study indicated that Cy5.5-labeled DXS-*b*-PCL nanoparticles were preferentially taken up by the stimulated macrophages in a receptor-mediated manner than by the non-stimulated macrophages. In vivo biodistribution studies demonstrated that these nanoparticles are selectively accumulated into the inflamed synovia of collagen-induced arthritis (CIA) mice due to SR-A mediated binding, compared to those of wild-type (WT) mice (Figure 6). These findings suggested that DXS-b-PCL nanoparticles have the potential as drug carriers for treating arthritis.

Besides DXS-based optical nanoprobe, a few DXS-based nanotheranostic agents have also been developed for combined optical imaging and therapy. For example, Park and co-workers prepared NIR fluorescence dye FPR-675-labeled self-assembled DXS nanoparticles loaded with anti-rheumatic drug methotrexate (MTX) as a theranostic platform for RA imaging and therapy [74]. These nanoparticles were prepared using an amphiphilic DXS conjugate, which was synthesized by chemically conjugating 5β-cholanic acid to the hydroxyl functional group of the DXS polymeric backbone. In vivo optical imaging after systemic administration into experimental CIA mice indicated about 12-fold enhanced accumulation of these DXS nanoparticles in inflamed joints compared to that of WT mice. Furthermore, these MTX-loaded DXS nanoformulation showed improved therapeutic efficacy against CIA in mice compared to free MTX. Recently, Song et al. synthesized a macrophage targetable NIR fluorescence emitting DXS-Chlorin e6 (DXS-Ce6) phototheranostic agent by covalently conjugating Ce6 to the DXS backbone and investigated its phototherapeutic feasibility in murine atheroma (Figure 7) [75]. As expected, owing to the SR-A mediated endocytosis, these DXS-Ce6 phototheranostic agents effectively internalized into the activated macrophages and foam cells. Image-guided photoactivation of DXS-Ce6 was able to detect in vivo inflammatory activity in atheroma and able to reduce both plaque burden and inflammation in murine models. A detailed immuno-fluorescence and histochemical analysis revealed that laser irradiation of DXS-Ce6 emits NIR fluorescence and concomitantly produces reactive oxygen species, which activates autophagy and upregulates MerTK expression within foam cells and subsequently promotes the engulfment of photoactivation-induced apoptotic cells. Thus, macrophage-targeted photoactivation of DXS-Ce6 reduces inflammatory activity and results in the regression of the inflamed plaque, indicating the potential of this phototheranostic agent for high-risk atheroma.

## 5. Conclusions

The excellent biocompatibility and intrinsic negative charge of DXS enable them to design and construct diverse nanocarriers for the delivery of a wide range of therapeutic and imaging agents. In several studies, DXS-based nano-sized drug delivery systems were prepared by simple complexation with either charged small molecule drugs or therapeutic macromolecules. Its ability to form polyelectrolyte nanocomplexes with polycations, specifically with CH, has been judiciously utilized to develop drug carriers for various therapeutic applications. Because of the mucoadhesive characteristics of DXS-CH nanocarriers, these systems have been widely employed for ocular drug delivery. Although PEC nanocomplexes between DXS and CH have been widely employed for the delivery of small molecule drugs to fragile proteins, the preparation of stable DXS-CH PEC nanoparticles requires optimization of various conditions such as molecular weights, degree of acetylation of CH, charge density and concentration of solutions, which is laborious and time consuming. Furthermore, to improve the stability of these nanoparticles, covalent crosslinking approaches have been utilized. Owing to the availability of about one hydroxyl group per repeating sugar unit in the DXS backbone, mostly chemical crosslinking is carried out at the CH via carbodiimide chemistry. Alternatively, the ability to chemically modify DXS allows the design of self-assembled nanoparticles or prodrug micelles for targeted delivery of poorly water-soluble drug molecules. In particular, the specific SR-A on activated macrophages facilitated DXS-based nanoparticles for imaging and therapy of RA. Despite these advantages, DXS-based nanoparticles were not explored much for various applications or have not entered clinical trials, such as its parental non-sulfated polymer dextran. Compared to dextran, only a few theranostic nanoparticles have been designed, prepared and investigated. Therefore, in the future, specific attention must be paid to the development of multifunctional theranostic nanoplatforms for the combined diagnosis and treatment of various diseases.

## Figures and Tables

**Figure 1 ijms-24-00355-f001:**
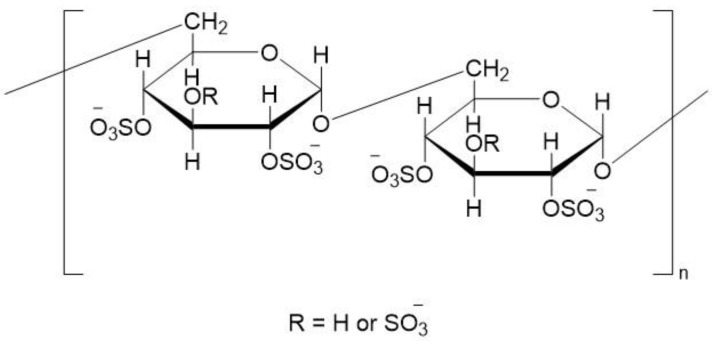
Chemical structure of DXS.

**Figure 2 ijms-24-00355-f002:**
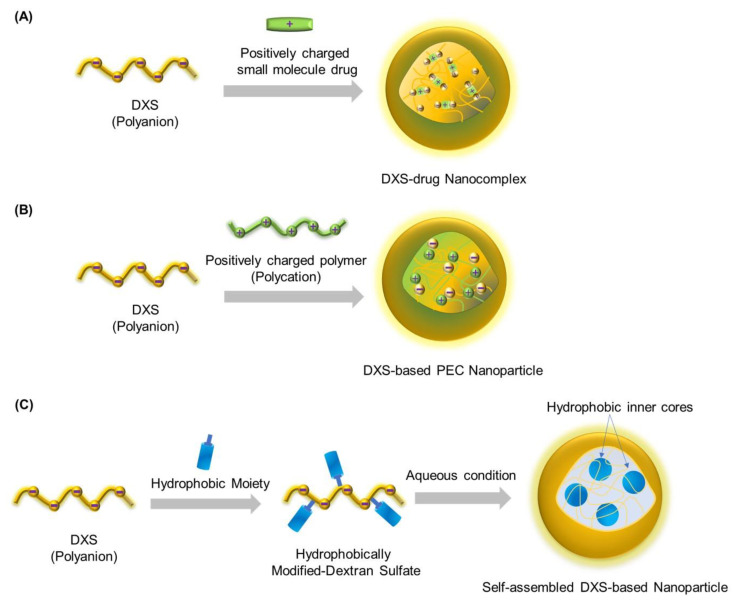
Schematic illustration for the preparation of DXS-based nanoparticles through various mechanisms. (**A**) complexation of DXS and positively charged small molecule drugs, (**B**) complexation of negatively charged DXS and positively charged polymers, and (**C**) self-assembly of hydrophobically modified amphiphilic DXS conjugate.

**Figure 3 ijms-24-00355-f003:**
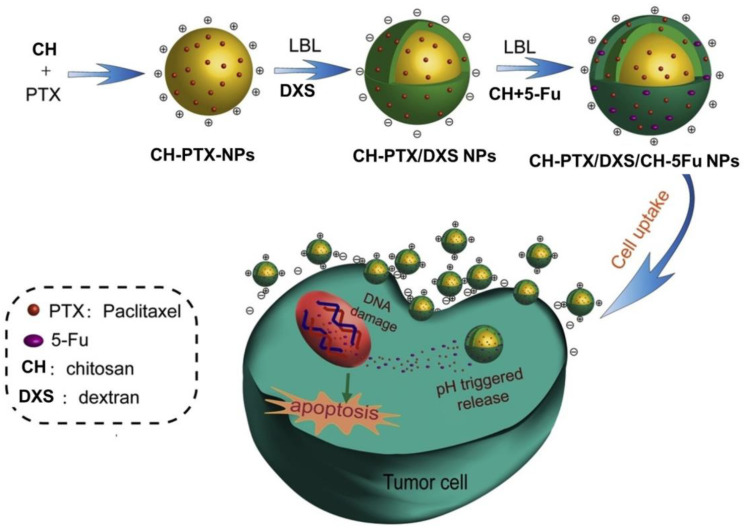
Schematic illustration of the preparation of dual-drug (PTX and 5-Fu)-loaded three layered CH/DEX/CH nanoparticles and cellular internalization and intracellular drug release. Adopted with permission from ref. [41] Copyright 2020 Elsevier B.V.

**Figure 4 ijms-24-00355-f004:**
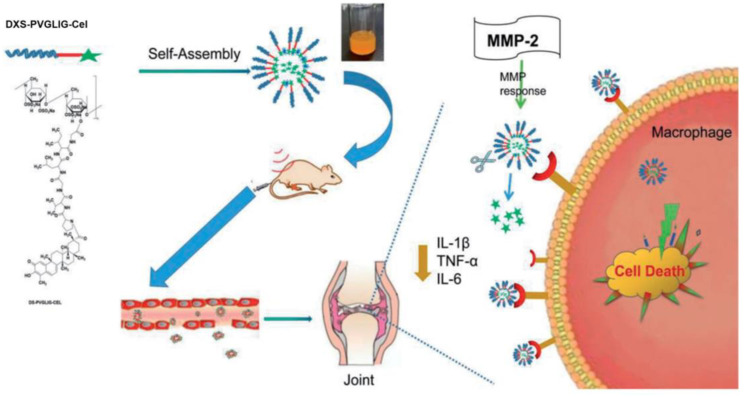
Schematic illustration for the preparation of MMP2-enzyme sensitive DXS-PVGLIG-Cel prodrug micelles and its SR-A mediated targeting for the treatment of RA. Adapted with permission from the ref [53]. Copyright 2022 The authors.

**Figure 5 ijms-24-00355-f005:**
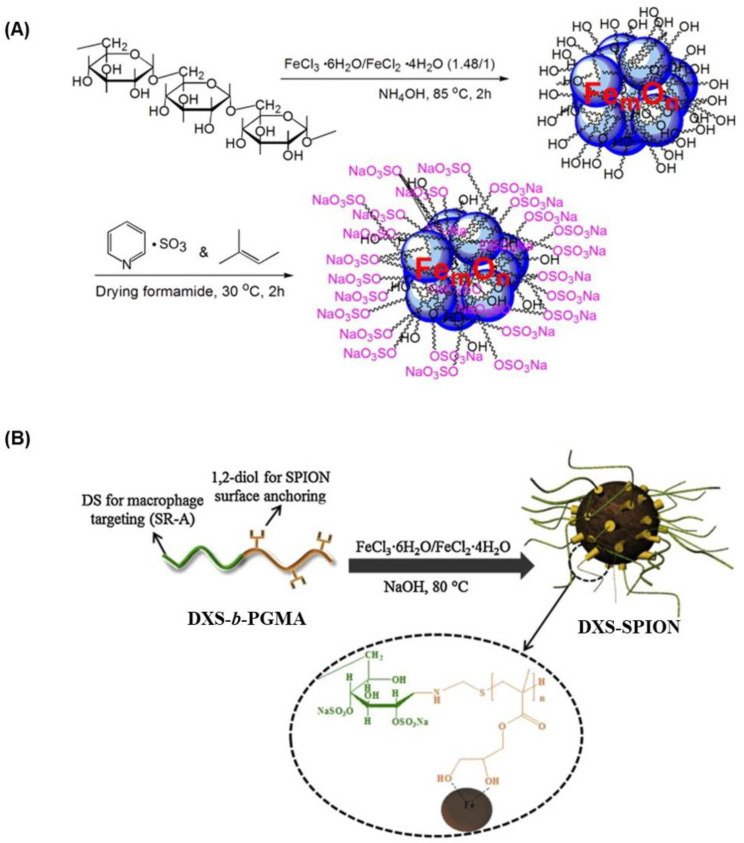
Preparation of DXS-coated SPIONs by various methods for MRI. (**A**) Sulfating dextran-modified SPIONs. Adapted with permission from ref [66] Copyright 2011 Elsevier Ltd., and (**B**) Coating SPIONs with double hydrophilic DEX-b-PGMA copolymer, Adapted with permission from ref [68] Copyright 2013 Elsevier Ltd.

**Figure 6 ijms-24-00355-f006:**
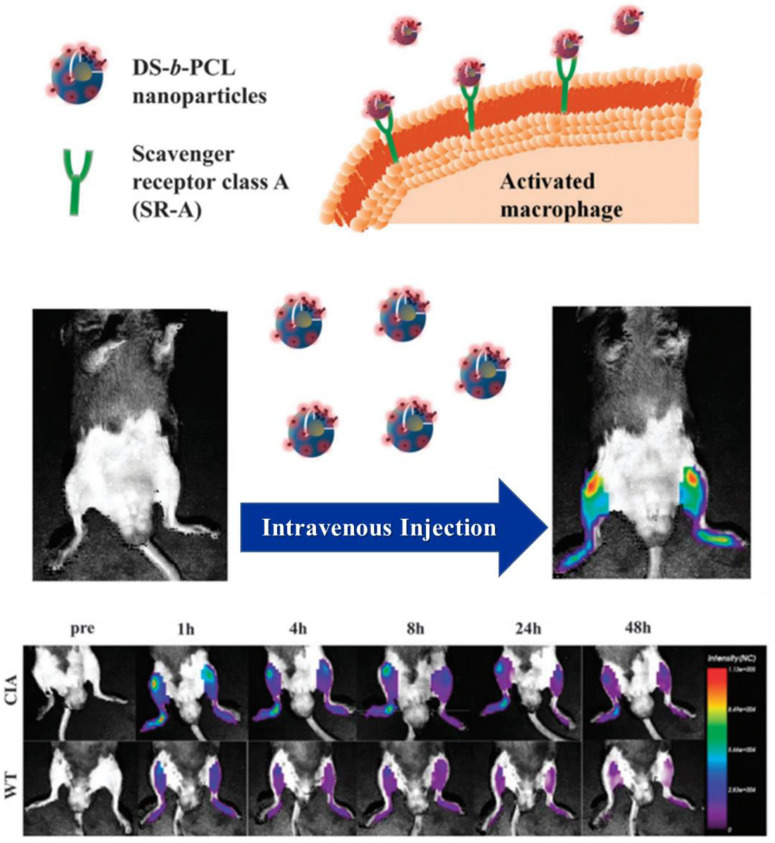
Schematic illustration of DXS-b-PCL nanoparticles for SR-A receptor-mediated targeted imaging of RA. In vivo biodistribution of the nanoparticles in CIA mice. Adapted with permission from ref [73]. Copyright 2013 The Royal Society of Chemistry.

**Figure 7 ijms-24-00355-f007:**
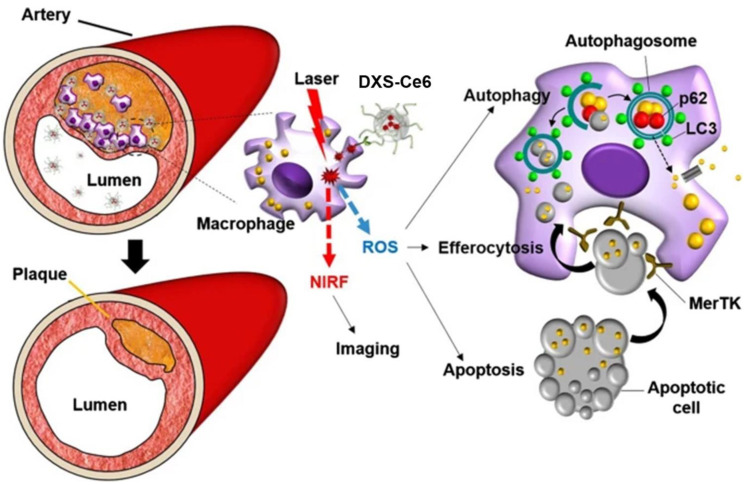
Schematic illustration of the macrophage SR-A targeted NIR fluorescence imaging and photoactivation of DXS-Ce6 for autophagy induction and efferocytosis enhancement to regress atherosclerosis. Adapted with permission from ref [75]. Copyright 2021 The authors.

## Data Availability

Upon reasonable request, the data supporting this investigation are available from the corresponding author.

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
