# Peer review of "Dextran Sulfate Nanocarriers: Design, Strategies and Biomedical Applications"

_ijms, 2022, doi:10.3390/ijms24010355_

Round 1

Reviewer 1 Report

The review article by Ramasundaram et al. summarizes recent progresses on an important and interesting delivery vector with great potentials for biomedical applications such as drug delivery and imaging. Overall, the article is well organized and the presentation in general is good. Some minor issues could be further improved to enhance the significance of the manuscript:

-One aspect that could be addressed/discussed is the stability of DXS nanoparticles for therapeutics applications, as vector stability both in vivo and in vitro are important issue for drug delivery. however, there are few studies addressing this important aspect.

-Has the DXS nanoparticles for drug delivery been tested in clinical trials?  if yes, these info could be added, and potential toxocoty/safety issues could be included.

-Cancer cell targeting 

The specificity/targeting capacity of DXS nanoparticles in cancer treatments could be summarized if there are studies on the issue, or it could be discussed as future perspectives. 

-DXS nanoparticles advantages and limitations 

In comparison to other types of nanoparticles, the key advantages and challenges in the production and applications of DXS nanoparticles could be discussed/summarized more in detail as this could offer the readers info about why or when to use DXS NPs but not other NPs.

-While the title of Fig. 1 is ‘Chemical structure of DXS’, and the chemical structure of sodium salt of DXS was shown. The title could be more precise.

Author Response

All the comments arisen from the reviewer were carefully studied and reflected in the revised manuscript. The revised parts are marked in blue. The authors appreciate reviewer’s valuable comments. 

Reviewer 2 Report

1. Many similar reviews have been published, and this article should focus on developments following those reviews. Many references are too old and should be minimized as much as possible.

2. The drawbacks of this material should also be discussed, which relate to the potential applications of the material.

3. Permission should be obtained from the journals.

4. Font size in fig 2 should be enlarged.

Author Response

(The authors gave the same response as above.)

Reviewer 3 Report

The author thoroughly discussed how DXS has been utilized as drug nanocarrier. I have a few comments below: 

"et al" should be "et al."

(page 2) The subsection name  "2. Strategies for the preparation of DXS nanoparticles" maybe inaccurate. Suggest"2.Strategies for the preparation of DXS-based nanoparticles" based on section 2.1 to 2.3. 

• The author mentioned cytotoxicity effect in Section 3.1 Cancer. Are there any other risks in Section 3.1 to 3.3?

Author Response

(The authors gave the same response as above.)
